# Multi-Scale Evaluation of the Effect of Thermal Modification on Chemical Components, Dimensional Stability, and Anti-Mildew Properties of Moso Bamboo

**DOI:** 10.3390/polym14214677

**Published:** 2022-11-02

**Authors:** Xiao Xiao, Xingyu Liang, Haozhe Peng, Kaili Wang, Xiaorong Liu, Yanjun Li

**Affiliations:** 1Jiangsu Co-Innovation Center of Efficient Processing and Utilization of Forest Resources, Nanjing Forestry University, Nanjing 210037, China; 2Bamboo Engineering and Technology Research Center, State Forestry and Grassland, Nanjing 210037, China

**Keywords:** bamboo, thermal modification, bamboo cell wall, anti-mildew property

## Abstract

By promoting greenhouse gas sequestration, bamboo and bamboo-based products can improve carbon storage, and thus help decrease greenhouses gas emission through replacing traditional products like concrete, steel, and alloy. Thermal modification is a useful way to effectively enhance the dimensional stability and mold-resistance property of bamboo and bamboo-based products compared with chemical treatment. This work investigates the change in anti-mildew properties, micro-structure, and chemical composition of bamboo after heat treatment. Saturated steam heat treatment was applied for this project. SEM results showed that the structural damage of parenchyma cells resulted in the separation of thin-walled cells and vascular bundles. Thus, the original regular structure of bamboo, characterized by plump and intact cells, changed markedly. After thermal modification, bamboo samples exhibited improved dimensional stability and anti-fungal properties due to the decrement of hemicellulose and cellulose. The hardness and MOE of the modified bamboo were 0.75 and 20.6 GPa, respectively.

## 1. Introduction

There are 78 genera divided into 1500 species of bamboo all over the world, and the area of bamboo forests is about 20 million square hectares. By promoting greenhouse gas sequestration, bamboo and bamboo-based products can improve carbon storage, and thus help decrease greenhouses gas emission through replacing traditional products like concrete, steel, and alloy [1,2,3,4]. In the past decades, bamboo has caused wide concern in the construction, building, decoration, and other fields due to its advantages such as excellent mechanical properties, easy harvesting, low density, and good safety [5]. When bamboo and bamboo-based products are applied in decoration and construction projects, they can be easily affected by fungi [6], water [7], and UV [8]. Therefore, it is of great important to find a modification method that can effectively improve the dimensional stability, physical properties, and mold-resistance property of bamboo, and thus expand the application field of bamboo and bamboo-based products and extend its service life [9,10,11].

Heat treatment is a useful way of effectively enhancing the dimensional stability and mold-resistance of bamboo-based products compared with that of chemical treatment [12]. In the thermal modification process, the bamboo becomes more dimensionally stable and less hygroscopic due to the decomposition of hemicelluloses, crystallinization of cellulose, remification of lignin, and removal of extractives. These changes can help bamboo [5] reduce moisture content re-absorption, and thus improve weathering and durability, and increase the dimensional stability [13]. More specifically, thermal modification can effectively improve the hygroscopicity, swelling, and shrinkage when the treatment temperature is over 150 °C. At 180 °C or higher, the anti-fungal property can be positively enhanced due to the decomposition of hemicellulose and cellulose [5].

Modern spectroscopic techniques, such as the Wet chemistry method, X-ray diffractometer, Fourier transform infrared spectroscopy, and anti-fungal test, are effectively methods for analyzing the relationship between thermal modification and the change in chemistry components [14]. Thus, the thermal modification mechanism can be deeply investigated through these surface spectroscopic techniques. Previous works focused on the changes in the surface chemical properties and the micro-morphology of woody resources after high-temperature heat treatment [15,16,17,18], ignoring the change in anti-mildew properties, dimensional stability, and micro-mechanical property [6]. Changes in micro-morphology, chemical composition, crystallinity index, anti-mildew properties, and dimensional stability of bamboo after heat treatment are still unclear. In addition, bamboo cell walls are composed of hemicellulose, cellulose, and lignin, and the relationship between chemical composition change and micro-mechanics have rarely been investigated [19]. Nanoindentation (NI) can directly reveal the response between the thermal modification parameters and micro-mechanical properties of bamboo cell walls [20]. Thus, it is meaningful to investigate the change in nano-mechanics of bamboo at the cell-wall level [21].

This work investigates the change in anti-mildew properties, micro-structure, and chemical components of bamboo after heat treatment. High temperature steam was applied for this project. We revealed the thermal modification mechanism of bamboo through the Wet chemistry method, X-ray diffractometer (XRD), Fourier transform infrared spectroscopy (FTIR), physical/mechanical properties test, and anti-fungal test.

## 2. Materials and Methods

### 2.1. Sample Preparation

Six-year-old moso bamboo was used in this work. Bamboo was harvested in JiangXi Provience, China. The arc-shaped bamboo sheets were 1050 mm × 11 mm × 120 mm, and the surfaces of the bamboo sheets were smooth and no defects. The bamboo sheets were directly transformed by heat treatment at different thermal modification treatment temperatures and the same durations (6 min, 8 min, and 10 min). Saturated steam was used as a thermal modification medium. Pressure tank (RDW-1.5-D, Rongda Boiler Container Co., Nanjing Ltd., Nanjing, China) was provided by Hangzhou Rongda Boiler Container Co., Ltd., Hangzhou, China.

### 2.2. Scanning Electron Microscopy

Bamboo specimens with average dimensions of 5 mm (length) × 5 (width) × 1 mm (thickness) were cut from different bamboo samples. The cross-section surface of different bamboo specimens were polished with a knife and coated with gold for observation by scanning electron microscope (Quanta 200, FEI, Tokyo, Japan).

### 2.3. XRD

The bamboo powders used in the Wet chemistry method were also used here. During the XRD test, the crystallinity degree of different bamboo specimens was determined by X-ray diffractometer (Ultima IV, Tokyo, Japan). The bamboo powders were exposed to X-ray radiation. The 2-theta and scan rate were set from 5° to 45° and 2 min^−1^, respectively. Segal’s method was used to calculate the crystallinity index [22].

### 2.4. FTIR

Fourier transform infrared (FTIR) spectroscopy was used to investigate the chemical reaction between bamboo and thermal modification parameters, using the VERTEX 80V Spectrum (Bruker, Berlin, Germany) at room temperature. Untreated and treated bamboo samples were milled to powders of 100 mesh and then pressed together with KBr powders into transparent film. The resolution of the device was 2 cm^−^^1^ with 32 accumulations. The range of the wavelength was recorded from 400 to 4000 cm^−^^1^.

### 2.5. EMC, ASE, and Mass Loss Test

The different bamboo samples were cut into small blocks with average dimensions of 20 × 20 × t mm (length × width × thickness). Then, all bamboo specimens were placed in an incubator (HWS-250, Jinhong Co., Ltd., Nanjing, China) under relative humidity of 65% and a temperature of 20 °C for 14 days. Lastly, the EMC, ASE, and mass loss of untreated and treated bamboo samples were tested according to the Chinese National Standard GB/T 15780–1995 “Testing methods for physical and mechanical properties of bamboo” [20].

### 2.6. Wet Chemistry Method

Before the chemical components analysis, the different bamboo samples were ground into powder and passed through a sieve with an average size of 40–80. Approximately 200–300 mg of the bamboo samples were placed in a tube, and 3 mL of 72% H_2_SO_4_ was added to submerge the bamboo specimens. The detailed experimental process can be found in National Renewable Energy Laboratory (NREL).

### 2.7. Anti-Mildew Property Test

The anti-mildew property of the different bamboo specimens was analyzed according to the National standard GB/T 18261-2000, “The method for control of wood mold and cyanobacteria by mildew inhibitor”, with Aspergillus niger as representative mold. In the national standard, one month is a complete measurement period.

## 3. Results and Discussion

### 3.1. Micro-Morphology Analysis

SEM was used to detect the micro-morphology of bamboo specimens. As seen from Figure 1, the treated bamboo samples exhibited a deformation of the thin-walled cell after thermal modification. The shape of parenchyma cells changed from “round and smooth” to “flat and distort” [23]. These changes indicated that the treated bamboo’s thin-walled cell became fragile and loose in comparison to the control [24]. This is because of the high-temperature steam inserted into bamboo inner tissue, which lead to the degradation of starch and hemicellulose [7,8,9,10]. The Wet chemistry method, XRD, FTIR, and physical tests are needed to further explore the effects of thermal modification on functional groups of bamboo samples.

### 3.2. Chemical Components and Mass Loss Ratio Analysis

In the Wet chemistry method test, the relative content of hemicellulose, cellulose, and lignin of the control were 21.5%, 41.5%, and 22.3%, respectively. Modified bamboo samples had a low hemicellulose and cellulose content and high lignin content, indicating that thermal modification can accelerate the decomposition of hemicellulose, which is conducive to the decomposition of polysaccharide and starch, allowing enhancement of mechanical and mold resistance. Especially compared to the decreasing tendency of hemicellulose and cellulose content, the lignin content exhibited an increasing tendency. According to previous research [11,12,13], the main composition of hemicellulose is xylan, which easily decomposes and dehydrates due to its branched and amorphous structure. Therefore, hemicellulose is easier to degrade than cellulose and lignin. At 160 °C or higher, the lignin content increases quickly with the increment of treatment temperature, which can be attributed to the lignin condensation reaction [14]. For detail, the polysaccharides in hemicellulose have low thermal stability under high temperatures due to their branched structure and amorphous structure, making it easier to decompose hemicellulose than other chemical components in bamboo [15]. The results of mass loss ratio are shown in Figure 2B. As shown in Figure 2B, with the increasing treatment temperature and duration, the mass loss increased. This is due to the decomposition of carbohydrate polymers in bamboo samples. Additionally, the decomposition of extractives can also contribute to this conclusion [16,17,18]. Figure 3 shows the starch content of different bamboo samples. The starch content of bamboo samples decreased from 3.25% to 2.90%. This is due to the high temperature and high pressure provided by saturated steam.

### 3.3. XRD and FTIR Analysis

Figure 4A shows the XRD patterns and CrI of different bamboo samples. In the high temperature-treated bamboo samples, the CrI increased from 40.5% to 59.5%. It is well known that biomass cellulose contains quasi crystalline regions, this being attributed to rearrangement or reorientation of cellulose molecules inside these regions. In addition, the degradation of cellulose in the amorphous region may happen at high temperatures, which results in more crystallization. Therefore, the crystallinity index of treated bamboo specimens increased. In addition, the decomposition of the para-crystalline part of cellulose can also make a positive contribution to the increment of cellulose crystallinity index [19,20,21,22].

As shown in (Figure 4C,D, the FTIR curves of untreated and treated bamboo specimens from 500 cm^−^^1^ to 4500 cm^−^^1^ were presented. The spectra of different bamboo samples presented the typical peaks of bamboo: 1731 cm^−^^1^ (C=O strength vibrations of hemicellulose), 1230 cm^−^^1^ (C-O strength vibrations peaks), 1590 cm^−^^1^ (stretching of carboxylic acid), and 1425 cm^−^^1^ (strengthening of acetyl acid). It can be seen from the (C) that there are no significant differences between the FTIR spectra of untreated and treated bamboo specimens. The relative intensities of peak at 1370 cm^−^^1^ do not change too much due to the stability of C-H, illustrating that C-H can remain unchanged during the thermal modification. The relative intensity of peaks at 1230 cm^−^^1^ and 1730 cm^−^^1^ show decreasing tendency in comparison to that of the control, which was due to the decomposition of the cellulose and hemicellulose in bamboo samples. During the saturated steam heat treatment process, the relative intensity of peaks at 1630 cm^−^^1^ and 1590 cm^−^^1^ increased with the increasing treatment temperature. This can be attributed to the increase in lignin content. The condensation reaction of lignin positively contributes to the increase in relative lignin content and can enhance the dimensional stability of the bamboo specimen. Lastly, the relative intensity of peak at 898 cm^−^^1^ obviously decreased, possibly resulted from acid environment provided by the decomposition of hemicellulose.

### 3.4. EMC and ASE Analysis

Figure 5A,B shows the EMC and ASE of the saturated steam treated bamboo samples. The EMC and ASE of the control were 13.5% and 8.4%, respectively. The EMC and ASE of treated bamboo specimens presented similar change stages. In Figure 5A, the EMC decreased with the increment of thermal modification parameters. In detail, we can find that the bamboo samples exhibited the lowest value (7.45%) of EMC under 180 °C and 10 min. These results suggest the degradation of extractives, ash, and starch in bamboo inner tissue and thus increases hygroscopicity of treated bamboo samples. This is due to the decrement of OH groups in hemicellulose [23,24,25,26,27].

### 3.5. Bamboo Cell Wall Mechanics

Modulus of elasticity and hardness are two measurement indexes to evaluate cell wall mechanics in Nanoindentation tests. The tested bamboo sample pictures nanoindentation curves were inserted in Figure 6A,B. Figure 6A,B shows the MOE and H of the saturated steam treated bamboo samples. The MOE and H of the control were 15.5 GPa and 0.59 GPa, respectively. The elastic modulus and hardness of the saturated steam-treated bamboo samples presented a similar increasing tendency. In Figure 6A, the MOE and H increased with the increment of thermal modification parameters. For example, the bamboo specimens (180 °C/10 min) exhibited highest MOE (20.6 GPa) compared with that of the control. At the same time, the hardness of the bamboo exhibited an increasing tendency, from 0.59 GPa to 0.75 GPa. The lignin condensation can endow the increment of bamboo cell wall mechanics. Additionally, the condensation polymerization and decreased EMC can also positively affect the bamboo cell wall mechanics [2,28,29,30,31,32,33].

### 3.6. Mold-Resistance Property Analysis

Figure 7A,B show the anti-mildew property and correspondeding pictures. In order to evaluate the mold-resistance property of the bamboo samples, the bamboo samples were incubated by *Aspergillus niger* for 30 days. It was observed that the infection ratio of the control was 0% in the first test day. After 8 days’ infection, the *Aspergillus niger* appeared in the bamboo surface, illustrating that the bamboo samples were incubated in a very short time. As shown in Figure 7A, after only 8 days’ incubation, the infection ratio of the control was 100%. However, the saturated steam-treated bamboo samples (180 °C and 10 min) exhibited better anti-mildew properties in comparison to the control. This is because starch and polysaccharide decreased in bamboo specimens after thermal modification. Additionally, the enhanced relative content of lignin in bamboo samples can inhibit the adhesion between *Aspergillus niger* and the bamboo surface, which can improve the anti-mildew property of bamboo samples [34,35,36,37,38,39,40].

## 4. Conclusions

By promoting greenhouse gas sequestration, bamboo and bamboo-based products can improve carbon storage, and thus help decrease greenhouses gas emission through replacing traditional products like concrete, steel, and alloy. In summary, a green and cost-effective method was proposed for bamboo thermal modification. The mold-resistance property of bamboo was enhanced due to the degradation of starch and hemicellulose. The MOE of the modified bamboo increased from 15.5 GPa to 20.6 GPa. At the same time, the hardness of the bamboo exhibited an increasing tendency, from 0.59 GPa to 0.75 GPa. Because of the degradation of hemicellulose, the dimensional stability of the modified bamboo was enhanced and the mass loss test confirmed the degradation of ash, chemical composition, and extractives in bamboo tissue. We believe that this work can help people focus on bamboo resources and deeply understand the thermal modification mechanisms.

## Figures and Tables

**Figure 1 polymers-14-04677-f001:**
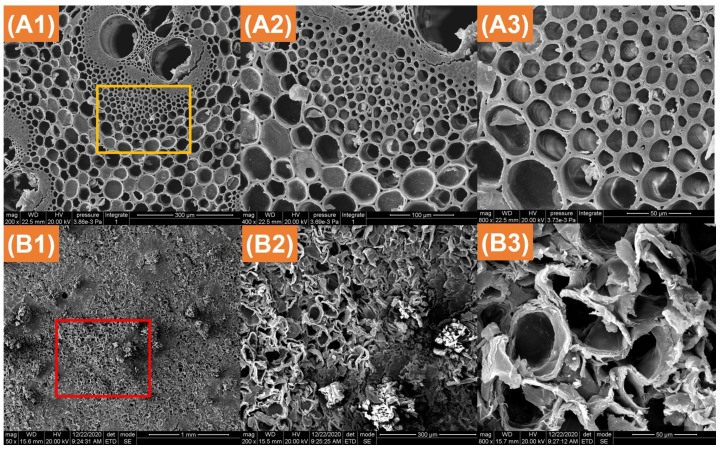
(**A1**–**A3**) Images of untreated bamboo in the cross-section and (**B1**–**B3**) thermal modified bamboo treated under 180 °C/10 min.

**Figure 2 polymers-14-04677-f002:**
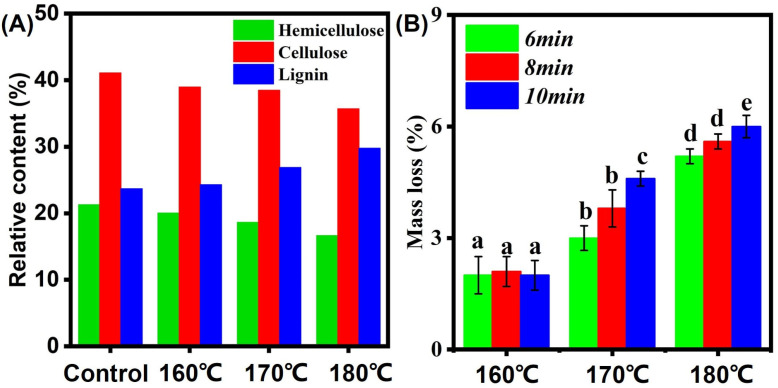
(**A**) Change in three major chemical compositions; (**B**) Mass loss. Different small (a–e) represent the significant difference between heat treatment groups. The error bar in the picture represents the standard deviation.

**Figure 3 polymers-14-04677-f003:**
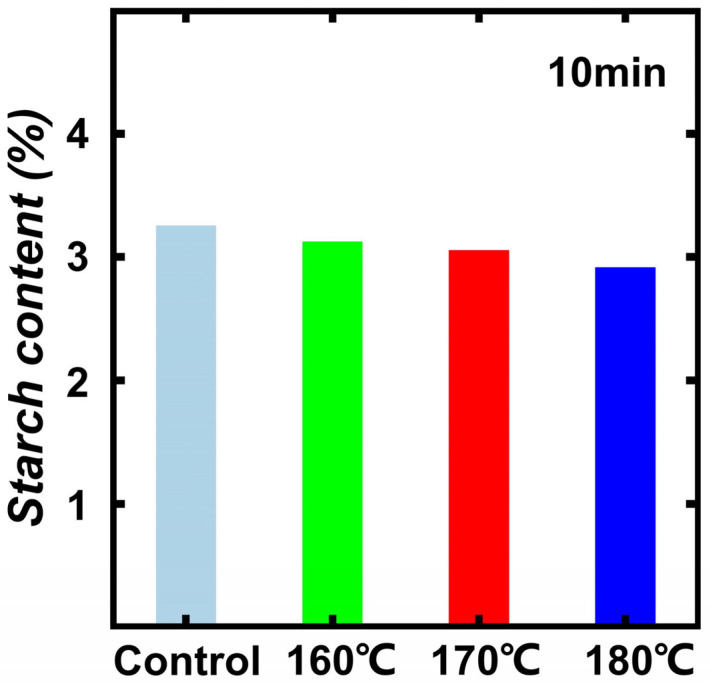
The starch content of different bamboo samples.

**Figure 4 polymers-14-04677-f004:**
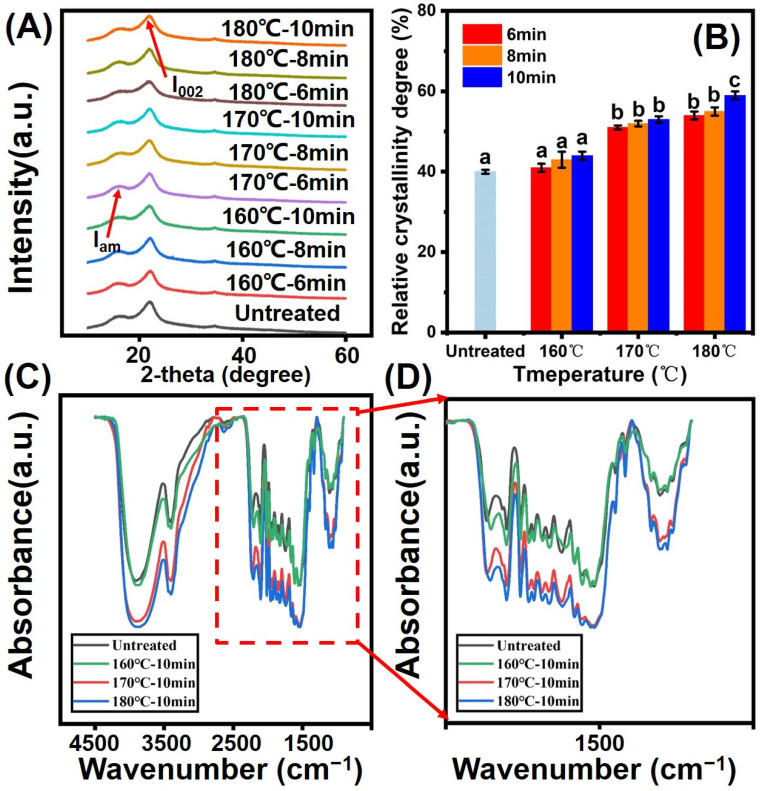
(**A**) XRD patterns of different specimens with the 2-theta angle ranges of 5–60; (**B**) Crystallinity index; (**C**,**D**) FTIR and enlarged FTIR curves. Different small letters shows significant differences between groups.

**Figure 5 polymers-14-04677-f005:**
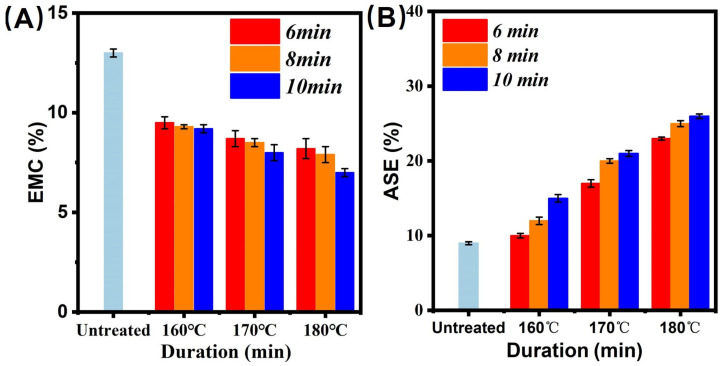
Physical property characterization. (**A**) EMC; (**B**) ASE.

**Figure 6 polymers-14-04677-f006:**
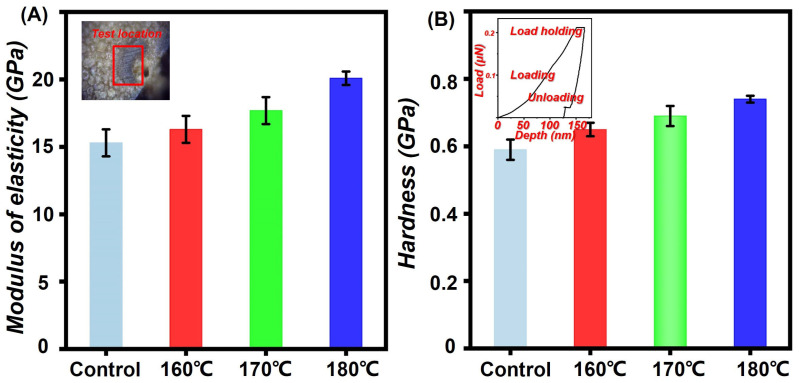
Micro-mechanical property characterization. (**A**) modulus of elasticity; (**B**) hardness.

**Figure 7 polymers-14-04677-f007:**
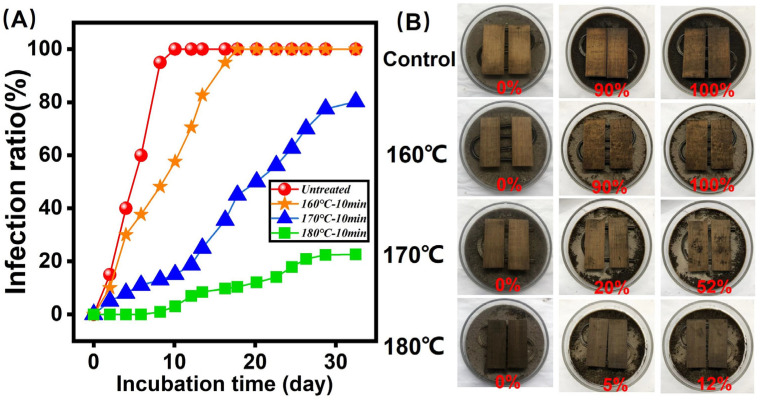
Micro-mechanical property characterization. (**A**) Infection ratio in one month; (**B**) Picture of anti-mold test in one month.

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
