# Peer review of "Multi-Scale Evaluation of the Effect of Thermal Modification on Chemical Components, Dimensional Stability, and Anti-Mildew Properties of Moso Bamboo"

_polymers, 2022, doi:10.3390/polym14214677_

Round 1

Reviewer 1 Report

Research article titled “Multi-scale evaluation of the effect of thermal modification on chemical components, dimensional stability, and anti-mildew properties of moso bamboo” submitted to Polymers, MDPI, by Xiao et al., is very interesting and well written. This paper is very timely and can be published without further revision. In addition, it is very suitable for wide readership of Polymers, MDPI.

 If possible, revise whole introduction section and abstract. Introduction section needs, clear objective and background of the work. Please see the following references (https://doi.org/10.1016/j.conbuildmat.2022.127320; https://doi.org/10.1016/j.pecs.2022.101023 ). These citations seem very necessary for this paper. 

Author Response

Reviewer 1:

Point 1: Research article titled “Multi-scale evaluation of the effect of thermal modification on chemical components, dimensional stability, and anti-mildew properties of moso bamboo” submitted to Polymers, MDPI, by Xiao et al., is very interesting and well written. This paper is very timely and can be published without further revision. In addition, it is very suitable for wide readership of Polymers, MDPI.

If possible, revise whole introduction section and abstract. Introduction section needs, clear objective and background of the work. Please see the following references (https://doi.org/10.1016/j.conbuildmat.2022.127320; https://doi.org/10.1016/j.pecs.2022.101023 ). These citations seem very necessary for this paper. 

Response: It is really a detail which should not be neglected as Reviewer suggested and a reference. We have revised whole introduction section and abstract. We added clear objective in our introduction section. We have readed the following references and cited it in our revised version. Thanks again to the reviewer on suggesting to properly address the significance of the work.

Reviewer 2 Report

I have read the manuscript, in general the manuscript needs to be improved for addressing some issues as follow 

1. In the introduction, the novelty of the research should be expressed more clearly. Why use the range of temperature in this study must be presented therefore can be informed what is the contribution of the research compared to previous research 

2. The potency of bamboo as a material source in this study should be presented in the data especially the abundance of Moso bamboo that used in this study 

3. In method must be presented in detail 

 - What kind of thermal pretreatment that applied in this study, what is the equipment used etc  

- The FTIR, XRD, sem also need informed in detail 

- In wet chemistry should be informed more, the step to do 

because in discussion inform about contribution extractive, ash and starch fore suggested presenting in method and data 

- The abbreviation should be informed previously 

4. Result and discussion 

-Many grammatical errors were found please check carefully such as temperature should be a degree in symbol, and the way to write Figure is a different way please check all 

- Add reference about condensation of lignin and add mechanism of it 

-line 100-101 : stated ...starch and hemicellulose but no data on starch also din some sentences about suggestion the contribution of ash and extractives but no data supported

- In XRD please give notification which one the crystalline and amorph in 2theta 

-line 130-131 : rearrangement of cellulose molecules, what is mean ? how is crystaline and amorphous part ? 

-What are MOE and H please give a description 

-Why use Aspergillus niger at 30 days in observation, please give an introduction reference 
5. Conclusion: need to revise based on the revised manuscript : such as the value of MOE AND h expressed only in one value 

6. Reference need to revise in the last publication in 10 year 

   see about the bamboo publication: https://doi.org/10.3390/polym14153111

Author Response

Reviewer 2:

Point 1: In the introduction, the novelty of the research should be expressed more clearly. Why use the range of temperature in this study must be presented therefore can be informed what is the contribution of the research compared to previous research 

Response: It is really a detail which should not be neglected as Reviewer suggested and a reference. We have revised the abstract and introduction to highlight the novelty of this work.

Point 2. The potency of bamboo as a material source in this study should be presented in the data especially the abundance of Moso bamboo that used in this study 

Response: It is really a detail which should not be neglected as Reviewer suggested and a reference. We have added the petency of bamboo as a meterials sourve in this study in the revised introduction section.

Point 3. In method must be presented in detail 

Response: It is really a detail which should not be neglected as Reviewer suggested and a reference. We have added the detailed method process in our revised version.

Point 4- What kind of thermal pretreatment that applied in this study, what is the equipment used etc  

Response: It is really a detail which should not be neglected as Reviewer suggested and a reference. We have added the kind of thermal pretreatment that applied in this study in the revised materials and methods section, and the equipment used in this manuscript.

Point 5: The FTIR, XRD, sem also need informed in detail 

Response: It is really a detail which should not be neglected as Reviewer suggested and a reference. The detailed FTIR and XRD information have been added into the revised manuscript.

Point 6: In wet chemistry should be informed more, the step to do 

Response: It is really a detail which should not be neglected as Reviewer suggested and a reference. We have added the detailed wet chemistry method in our revised version.

Point 7: because in discussion inform about contribution extractive, ash and starch fore suggested presenting in method and data 

Response: It is really a detail which should not be neglected as Reviewer suggested and a reference. In the wet chemistry method, ash and extractive can not be calculated in detail. Additionally, the starch are presented in data in our revised version.

Point 8: The abbreviation should be informed previously 

Response: Thanks again to the reviewer on suggesting to properly address the significance of the work. We have informed the abbreviation in our revised version.

  1. 4. Result and discussion 

Point 9: Many grammatical errors were found please check carefully such as temperature should be a degree in symbol, and the way to write Figure is a different way please check all 

Response: Thanks again to the reviewer on suggesting to properly address the significance of the work. We have revised the temperature in a symbol.

Point 10: Add reference about condensation of lignin and add mechanism of it 

Response: Thanks again to the reviewer on suggesting to properly address the significance of the work. We have added reference and add mechanism of it in our revised version.

Point 11:line 100-101 : stated ...starch and hemicellulose but no data on starch also din some sentences about suggestion the contribution of ash and extractives but no data supported

Response: Thanks again to the reviewer on suggesting to properly address the significance of the work. We have shown starch data in our revised version.

Point 12: In XRD please give notification which one the crystalline and amorph in 2theta 

Response: Thanks again to the reviewer on suggesting to properly address the significance of the work. I002 represented the crystalline in 2 – theta, Iam represented the amorph in 2-theta, we have revised that in our revised version.

Point 13:line 130-131 : rearrangement of cellulose molecules, what is mean ? how is crystaline and amorphous part ? 

Response: Thanks again to the reviewer on suggesting to properly address the significance of the work. We have revise it in our revised version. I002 represented the crystalline in 2 – theta, Iam represented the amorph in 2-theta, we have revised that in our revised version.

Point 14:What are MOE and H please give a description 

Response: Thanks again to the reviewer on suggesting to properly address the significance of the work. We have added some description of MOE and H in our revised version.

Point 15:Why use Aspergillus niger at 30 days in observation, please give an introduction reference 

Response: Thanks again to the reviewer on suggesting to properly address the significance of the work. The anti-mildew property of the different bamboo specimens were analyzed according to the National standard GB/T 18261-2000, “The method for control of wood mold and cyanobacteria by mildew inhibitor”, with Aspergillus niger as representative mold. In the nation standard, one month is a complete measurement period.

Point 16: Conclusion: need to revise based on the revised manuscript : such as the value of MOE AND h expressed only in one value 

Response: Thanks again to the reviewer on suggesting to properly address the significance of the work. We apoloized for this mistakes. We have revised it in our revised version.

Point 17: Reference need to revise in the last publication in 10 year 

   see about the bamboo publication: https://doi.org/10.3390/polym14153111

Response: Thanks again to the reviewer on suggesting to properly address the significance of the work. We have added this bamboo publication in our revised version.

Reviewer 3 Report

The article "Multi-scale evaluation of the effect of thermal modification on chemical components, physical/mechanical, and anti-mildew properties of moso bamboo" is devoted to the interesting topic of turning a national type of raw material into a valuable building material. Moreover, the procedure for processing raw bamboo is not too complicated. But the result is very curious and scientifically substantiated by the authors. Some roughness can be found in the text, but they will be eliminated during the editing process. As a future reader, thank you for the quality drawings.

Author Response

Reviewer 3:

Point 1:The article "Multi-scale evaluation of the effect of thermal modification on chemical components, physical/mechanical, and anti-mildew properties of moso bamboo" is devoted to the interesting topic of turning a national type of raw material into a valuable building material. Moreover, the procedure for processing raw bamboo is not too complicated. But the result is very curious and scientifically substantiated by the authors. Some roughness can be found in the text, but they will be eliminated during the editing process. As a future reader, thank you for the quality drawings.

Response: Thanks again to the reviewer on suggesting to properly address the significance of the work. We have revised some roughness in our revised text. It is really a detail which should not be neglected as Reviewer suggested and a reference. In addition, We have revised whole introduction section and abstract. We added clear objective in our introduction section.

Reviewer 4 Report

Following an overall inquiry into the reviewed article, I consider it to be a very interesting investigation of the multi-scale evaluation of the effect of thermal modification on chemical components, physical/mechanical, and anti-mildew properties of moso bamboo.

The manuscript contains interesting data, which have been mostly correctly evaluated and interpreted. Organization and clarity of the manuscript is also generally good. The paper resolves an elaborate multidisciplinary topic and meets formal layout standards and default criteria, imposed on such articles.

Thereby, I recommend its issuance.

I would like to ask the authors to marginally mention the following article in the introduction: Viglašová E. and Daňo M. et al.: (o) Engineered biochar as a tool for nitrogen pollutants removal: Preparation, characterization and sorption study. Desalin. Water Treat. 191: 318-331 (2020). (oo) Pertechnetate/perrhenate surface complexation on bamboo engineered biochar. Materials 14(3): 486 (2021).

Author Response

Reviewer 4:

Following an overall inquiry into the reviewed article, I consider it to be a very interesting investigation of the multi-scale evaluation of the effect of thermal modification on chemical components, physical/mechanical, and anti-mildew properties of moso bamboo.

The manuscript contains interesting data, which have been mostly correctly evaluated and interpreted. Organization and clarity of the manuscript is also generally good. The paper resolves an elaborate multidisciplinary topic and meets formal layout standards and default criteria, imposed on such articles.

Thereby, I recommend its issuance.

I would like to ask the authors to marginally mention the following article in the introduction: Viglašová E. and Daňo M. et al.: (o) Engineered biochar as a tool for nitrogen pollutants removal: Preparation, characterization and sorption study. Desalin. Water Treat. 191: 318-331 (2020). (oo) Pertechnetate/perrhenate surface complexation on bamboo engineered biochar. Materials 14(3): 486 (2021).

Response: It is really a detail which should not be neglected as Reviewer suggested and a reference. We have revised whole introduction section and abstract. We added clear objective in our introduction section. We have readed the following references and cited it in our revised version. Thanks again to the reviewer on suggesting to properly address the significance of the work.

Round 2

Reviewer 2 Report

Recommended to accept in current form